# Single-cell sequencing reveals the landscape of the human brain metastatic microenvironment

Qianqian Song [1], Jimmy Ruiz [2,3✉], Fei Xing [1], Hui-Wen Lo[4], Lou Craddock[1], Ashok K. Pullikuth [1], Lance D. Miller[1], Michael H. Soike[5], Stacey S. O'Neill[6], Kounosuke Watabe [1], Michael D. Chan [7✉] & Jing Su [8✉]

Brain metastases is the most common intracranial tumor and account for approximately 20% of all systematic cancer cases. It is a leading cause of death in advanced-stage cancer, resulting in a five-year overall survival rate below 10%. Therefore, there is a critical need to identify effective biomarkers that can support frequent surveillance and promote efficient drug guidance in brain metastasis. Recently, the remarkable breakthroughs in single-cell RNA-sequencing (scRNA-seq) technology have advanced our insights into the tumor microenvironment (TME) at single-cell resolution, which offers the potential to unravel the metastasis-related cellular crosstalk and provides the potential for improving therapeutic effects mediated by multifaceted cellular interactions within TME. In this study, we have applied scRNA-seq and profiled 10,896 cells collected from five brain tumor tissue samples originating from breast and lung cancers. Our analysis reveals the presence of various intratumoral components, including tumor cells, fibroblasts, myeloid cells, stromal cells expressing neural stem cell markers, as well as minor populations of oligodendrocytes and T cells. Interestingly, distinct cellular compositions are observed across different samples, indicating the influence of diverse cellular interactions on the infiltration patterns within the TME. Importantly, we identify tumor-associated fibroblasts in both our in-house dataset and external scRNA-seq datasets. These fibroblasts exhibit high expression of type I collagen genes, dominate cell-cell interactions within the TME via the type I collagen signaling axis, and facilitate the remodeling of the TME to a collagen-I-rich extracellular matrix similar to the original TME at primary sites. Additionally, we observe M1 activation in native microglial cells and infiltrated macrophages, which may contribute to a proinflammatory TME and the upregulation of collagen type I expression in fibroblasts. Furthermore, tumor cell-specific receptors exhibit a significant association with patient survival in both brain metastasis and native glioblastoma cases. Taken together, our comprehensive analyses identify type I collagen-secreting tumor-associated fibroblasts as key mediators in metastatic brain tumors and uncover tumor receptors that are potentially associated with patient survival. These discoveries provide potential biomarkers for effective therapeutic targets and intervention strategies.

[1] Department of Cancer Biology, Wake Forest University School of Medicine, Winston-Salem, NC, USA. [2] Hematology & Oncology, Department of Medicine, Wake Forest University School of Medicine, Winston-Salem, NC, USA. [3] W.G. (Bill) Hefner Department of Veteran Affairs Medical Center, Salisbury, NC, USA. [4] Department of Neurosurgery, McGovern Medical School, University of Texas Health Science Center at Houston, Houston, TX, USA. [5] Hazlerig-Salter Department of Radiation Oncology, University of Alabama-Birmingham Heersink School of Medicine, Birmingham, AL, USA. [6] Department of Pathology, Wake Forest University School of Medicine, Winston-Salem, NC, USA. [7] Department of Radiation Oncology, Wake Forest University School of Medicine, Winston-Salem, NC, USA. [8] Department of Biostatistics and Health Data Science, Indiana University School of Medicine, Indianapolis, IN, USA. ✉email: jruiz@wakehealth.edu; mchan@wakehealth.edu; su1@iu.edu

Brain metastasis refers to the malignant tumors that metastasize to the brain. It is estimated that brain metastasis occurs in at least 6% of all newly diagnosed cancer cases[1–3], and is a devastating cancer complication for the 200,000 patients each year diagnosed in the US[4]. The majority of brain metastases are from primary cancers such as lung, breast, and melanoma[5]. Brain metastasis are 10 times more common than primary brain tumors[6] and is associated with poor survival outcomes[6]. Substantial advances have been made in the diagnosis and treatment of patients with metastatic brain tumors, including targeted therapies that penetrate blood brain barrier and stereotactic radiosurgery as primary strategies of therapy[7]. Unfortunately, the mortality rate and the recurrence of brain metastasis remain high[8]. Therefore, there remains a need for effective biomarkers and potential targets to improve the diagnosis and treatment of patients with brain metastasis.

Accumulating evidence, including our work on multi-omics subtyping analysis, suggests that the tumor environment (TME) plays a major role in brain metastasis patients' responses to stereotactic radiosurgery and whole-brain radiation therapy[9]. Despite the difference in the origins of primary cancers, anatomic locations, and treatment histories, our recent work suggests that pan-brain metastasis biomarkers such as COL1A1/COL1A2[10] and CD37/cystatin A/IL-23A[11] are strongly associated with responses to radiotherapies, overall survival, and distant brain failure. Moreover, collagen genes including *COL1A1/COL1A2* are associated with *HER2* expression in patients with breast cancer brain metastasis[12]. Our previous work further suggests the intercellular communications within the tumor microenvironment may be responsible for the observed difference in the risk of distal brain failure. However, the molecular and cellular underpinnings of the immune landscape in brain metastatic cases are unclear[10].

The remarkable breakthrough in single-cell RNA-sequencing (scRNA-seq) technology advances the interpretation of the TME at single-cell resolution[13–15]. For example, Kim et al. reveal the stromal and immune cell dynamics that creates a pro-tumoral and immunosuppressive microenvironment in lung adenocarcinoma[16]. Chen et al. identify the inflammatory tumor-associated fibroblasts playing important roles in tumor progression and are significantly related to poor prognosis of bladder urothelial carcinoma[17]. Wu et al. identify large heterogeneity in cellular composition and rare cell types including follicular dendritic cells and T helper 17 cells in non-small cell lung cancer samples[18]. These studies highlight the significance of single-cell technology in accelerating the interrogations of TME and illuminating the causes and underlying mechanisms of human diseases especially cancers. Moreover, scRNA-seq technology is increasingly used to provide deep insights into complex cellular communications and the involved biological processes. As tumor ecosystem is composed of multiple cell types that communicate by ligand-receptor interactions, targeting ligand-receptor interactions may provide potential benefits for patients. However, our knowledge of which interactions occur in tumors and how these interactions affect tumor progression is still limited. Thus, the fast advance in single-cell technologies provides the potential for revealing intercellular communications within TME.

Though scRNA-seq shows a profound impact in primary tumors[19], similar studies in brain metastases are very rare, largely due to the difficulties in collecting clinical samples from brain metastasis. In this study, to comprehensively chart the cell landscape of patients with brain tumor metastasis, we apply scRNA-seq and generate high-resolution single-cell profiles from five patient samples with brain metastasis. Our analysis identifies intratumoral components with different cell types, as well as the tumor-associated fibroblasts, which highly express type I collagen and demonstrate enriched collagen-related signaling pathways. Further interrogation of cellular interactions within TME reveals the essential role of this identified tumor-associated fibroblasts. Such observations are also identified in external scRNA-seq datasets. Moreover, we reveal that the active collagen binding proteins expressed in tumor cells are indicative of prognosis in not only patients with brain metastasis of solid tumors, but also in patients with glioblastoma. We further explore the potential mechanisms and reveal that the M1 activation of native microglial cells and infiltrated macrophages create a proinflammatory microenvironment. Such wound-healing mechanism may be hijacked by metastatic tumor cells, which triggers excess expression of type I collagen, remodels the brain tissue into a microenvironment similar to the original TME at the primary cancer sites, and thus provides familiar niches for the metastatic tumor cells. Overall, our analyses identify the collagen I positive tumor-associated fibroblast as a key mediator in metastatic brain tumors and reveal the involved ligand-receptor signaling axis, which provides potential predictive biomarkers that can improve future strategies for surveillance, diagnosis, and effective treatments.

## Results

**Patient characteristics.** Patients' demographical and clinical characteristics are demonstrated in Table 1. Five brain metastatic tissues are analyzed in this study, including three from breast primary and two from lung primary. The five patients are 41–63 years old (median age: 55 years), all female, and majority white with one African American patient. Two of the three breast cancer brain metastases samples have primary triple-negative breast cancers, while the other one shows ER (estrogen receptor)/PR (progesterone receptor)/HER2 (human epidermal growth factor receptor 2) positive subtypes according the immunohistochemistry (for ER and PR) and in situ hybridization (for HER2). One of the lung cancer brain metastases samples has been examined without rearrangements of ALK and ROS1 and EGFR mutation. Two of the primary cancers metastasize to the left frontal lobe and three of the other primary tumors metastasize to cerebellum.

**Single-cell RNA sequencing and cell-type identification.** As shown in Fig. 1, after quality control and removal of the batch effect between batches, 10,896 single cells are retained for clustering analysis. The overall cell layout of the five samples is visualized using t-distributed stochastic neighbor embedding (t-SNE) plot (Fig. 1a), with cells from the same sample labeled with the same color. After batch effects removal, no obvious batch effect is observed, as cells from different samples are well-mixed. Cells are clustered into 17 major clusters to identify different cell types (Fig. 1b).

Classic signatures described in previous studies are used to annotate cell types. These signature gene expression patterns suggest 7 major cell types (Fig. 1c, d). Specifically, as tumor cells highly express epithelial features, typical epithelial markers including *KRT8* and *EPCAM* are highly expressed in 6 clusters (0, 3, 6, 9, and 10). Endothelial cells or pericytes (cluster 16), highly express hallmark genes in the blood-brain barrier[20,21], including *VWF*, *CLDN5* (claudin-5, major constitutive claudin at the blood-brain barrier), *CDH5*, and *ESAM*. Clusters 1, 12 and 13 are annotated as myeloid cells such as the brain-native microglia's and infiltrated peripheral macrophages for highly expressing *MRC1*, *CD163*, *TREM2*, *MSR1*, and *OLR1*. Cluster 8 is shown as T cells for highly expressing T-cell markers (*CD3*, *CD28*, *CD2*, *TRBC1*, *TRAC*, and *SKAP1*). Clusters 11 and 15 are annotated as oligodendrocytes with oligodendrocyte features including *GAL3ST1* and *GJC2*. Clusters 4 and 7 are classified as NSC

**Table 1 Patient and sample characteristics.**

| Characteristics | Samples | | | | |
|---|---|---|---|---|---|
| | BRBMET2 | BRBMET3 | BRBMET 87 | LUBMET7 | LUBMET1 |
| Patient | | | | | |
| Age | 41 | 55 | 56 | 63 | 46 |
| Gender | F | F | F | F | F |
| Race, n (%) | B | W | W | W | W |
| Pathology | | | | | |
| Tissue Type | Tumor | Tumor | Tumor | Tumor | Adjacent Tumor |
| Primary Tumor | | | | | |
| Site | Breast | Breast | Breast | Lung | Lung |
| Subtype | Invasive ductal carcinoma, Nottingham grade 3 | Unknown* | Invasive ductal carcinoma with micropapillary features, Nottingham grade 3 | Not sampled | Adenocarcinoma |
| Molecular Feature | ER negative, PR negative, HER2 negative by IHC | ER negative, PR negative, HER2 negative* | ER positive (2%; weak intensity), PR positive (5%; strong intensity), HER2 amplification by FISH | Not sampled | Negative for rearrangements of ALK and ROS1 by FISH; EGFR mutation negative |
| Brain Metastasis | | | | | |
| Site | Left occipital lobe | Cerebellum | Left cerebellum | Left frontal lobe | Cerebellum |
| Characteristics | ER negative, PR negative, HER2 IHC 2+; Amplification of Her2 detected by FISH | ER negative PR negative, HER2 negative by IHC | Micropapillary type; ER positive (2%; weak to intermediate intensity), PR negative, HER2 3+ by IHC | Not applicable | Poorly differentiated adenocarcinoma; ROS1 rearrangement positive by FISH; ALK FISH negative, EGFR mutation negative |

Samples from five patients diagnosed with brain metastasis of breast and lung origins are included. Fresh remnant tumor tissues are collected for single-cell RNA-sequencing.
ER estrogen receptor, PR progesterone receptor, HER2 human epidermal growth factor receptor 2, IHC immunohistochemistry, FISH fluorescence in situ hybridization.

(neural stem cell)-like stromal cells, based on the overexpression of NSC features such as *AURKB*, *CDCA3*, *UBE2C*, and *TK1*. Meanwhile, as clusters 2 and 5 do not present any over-expression of genes, these two clusters were classified as unknown. Of note, fibroblasts (cluster 14) are identified, with strong expression of hallmark genes such as type I collagens (*COL1A1* and *COL1A2*).

Myeloid cells (native microglial cells and infiltrated peripheral macrophages) are featured with hallmark genes of M1 activation[22]. For example, the major myeloid cell populations (clusters 1, 12, and 13) highly expresses interleukin 1β (*IL1B*), with the log-fold change (LFC) of 1.31–2.54 (Supplementary Data 1). Tumor necrosis factors (*TNF*), Transforming growth factor 1β (*TGFB1*), and Transforming growth factor β-induced (*TGFBI*) are also observed with over-expression. Chemokines such as *CCL5* (LFC: 0.84), *CCL3*, and *CCL4* (LFC: 3.17 and 3.27) are shown with elevated expressions in myeloid cells. Therefore, these myeloid cells in brain metastatic TME demonstrate the typical M1 activation gene expression pattern with pro-inflammatory functions.

Cell composition analysis of the 5 samples (Fig. 1e) shows substantial variation in the abundance of different cell types. While tumor cells and myeloid cells are consistently identified across patient samples, their relative proportions vary from patient to patient. For example, the proportion of oligodendro-cytes in a lung cancer brain metastasis sample (i.e., LUBMET1) was higher than in other samples. Proportional variances are also observed for other cell types. Overall, we found a large degree of variation in the cellular composition among the five specimens. Such phenotypic variability has also been reported in multiple cancer types[23–25], including brain metastatic tissues[26].

**Inference of cell–cell communications in TME.** Given the var-iations of different cell type compositions within TME, it may be due to the alteration of the complex intercellular communica-tions. Herein, we use the CellChat[27] tool, which is proposed to quantitatively infer the intercellular communication based on scRNA-seq data, to identify the cell-cell communications and how this cellular crosstalk may be relevant to protumor activities.

We first infer the numbers and strength of interactions among different cell populations in brain metastasis samples (Fig. 2a). Most interactions are observed among fibroblasts, tumor cells, NSC-like stroma, myeloid cells, and endothelial cells. Once the interaction strength is considered (represented by the interaction weights), fibroblasts demonstrate a pivot role in the tumor microenvironment, which strongly interacts with tumor cells, NSC-like stromal cells, myeloid cells, and endothelial cells. The inferred incoming and outcoming interaction strengths are elaborated in Fig. 2b, where fibroblasts are the major cell type expressing ligands and receptors that are actively involved in the cellular interactions. These results suggest that these fibroblasts are tumor associated and play a central role in the TME. To reveal the signaling pathways contributing to the complex intercellular communications, we calculate the outgoing and incoming interaction strength of each signaling pathway (Fig. 2c). Notably, collagen signaling is shown as the most predominant signaling in fibroblasts, as reflected by the largest outgoing and incoming interaction strength compared to other signaling pathways. In addition to collagen signaling, we also observe other increased extracellular matrix (ECM) signaling[28,29] related to fibronectin 1 (FN1) and laminin (LAMININ), as well as proteins related with brain injury and repairing[30] such as midkine (MK), pleiotrophin (PTN), and amyloid-beta precursor protein (APP). Consistent with these observations, the inferred cell-cell communication networks of the collagen signaling pathway reveal that collagen signaling is the core of the observed interactions between fibroblasts and other cells (Fig. 2d, e). In summary, collagen-based cell–cell interactions are the molecular underpinnings of the observed fibroblast-centered TME in brain metastatic tissues.

**Characterization of tumor-associated fibroblast cells.** High-lighted by the cellular crosstalk, fibroblasts are shown as an important player during brain metastasis. To further characterize this fibroblast cell population, we perform the differential expression analysis comparing fibroblasts with the rest of the cells. In Fig. 3a, we have identified 248 upregulated genes and 46 downregulated genes, including the hallmark fibroblast genes such as *DCN* and *LUM*, with the LFC of 2.68 and 1.64, respec-tively. Of note, *COL1A1* and *COL1A2*, related to type I collagen

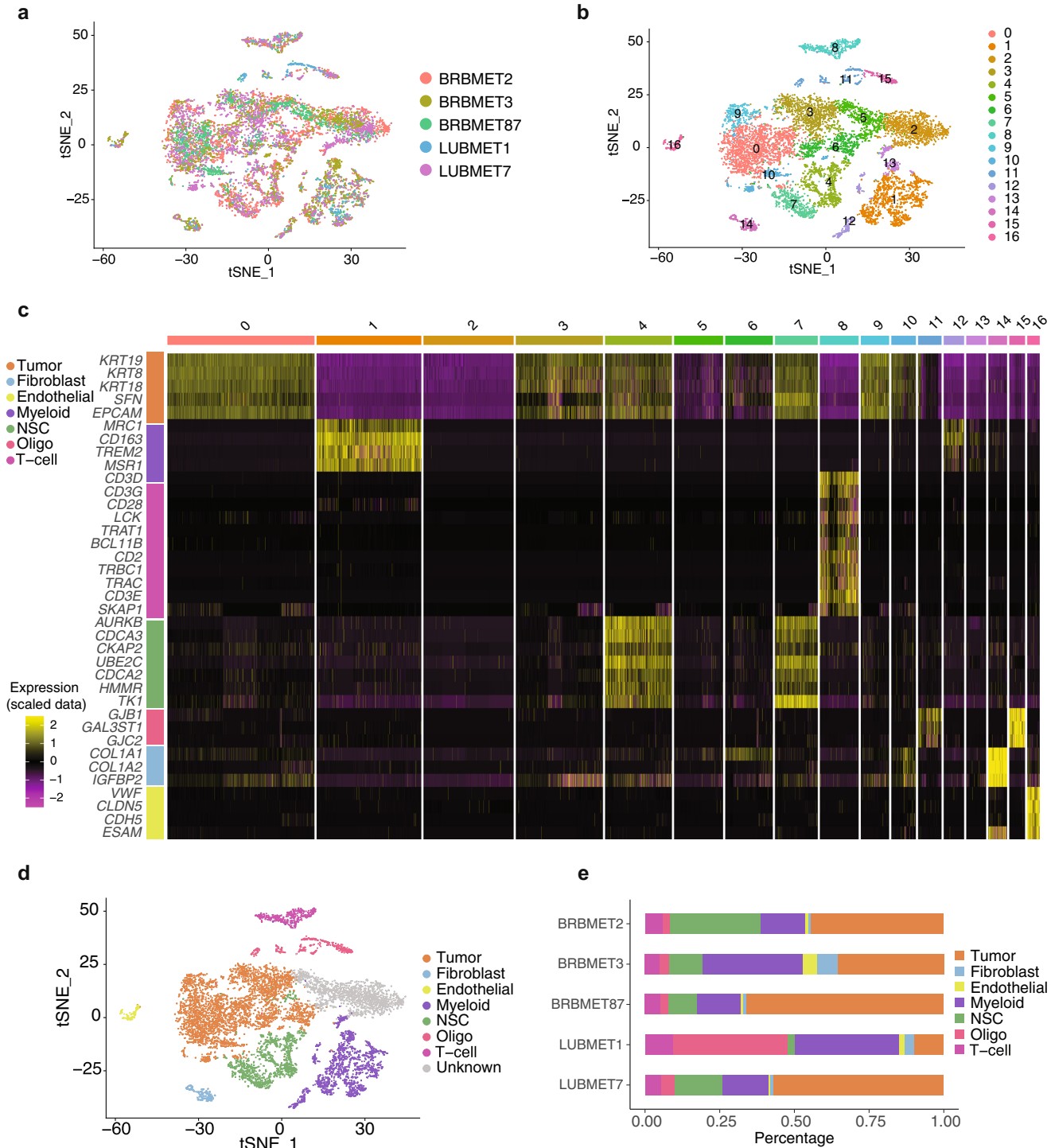

**Fig. 1 Single-cell RNA-seq data analysis and cell annotation. a** t-SNE plot of the profiled single cells. Cells with the same colors are from the same patient specimen. **b** t-SNE plot of the identified cell clusters. Different colors represent different cell clusters. **c** The heatmap shows the gene signatures of each annotated cell type. **d** t-SNE plot of the annotated cell types. Different colors represent different cell types. **e** Cell compositions of annotated cell types in each brain metastasis specimen.

signaling, are intensively expressed in fibroblasts (Fig. 3b, with LFCs of 3.58 and 3.20). Other collagen genes such as *COL3A1*, *COL4A1*, *COL4A2*, *COL6A1*, *COL6A2*, *COL18A1* are also highly expressed (with LFC > 2, Fig. 3a and Supplementary Data 1). The expressions of *SLC1A2* and *SLC1A3*, which encodes the glutamate transporters GLT-1 (*SLC1A2*) and GLAST (*SLC1A3*), are widely distributed throughout the brain[31], with some of them highly presented in fibroblast cells.

In addition, high expressions of tumor-related genes are observed in these fibroblasts. Notably, *IGFBP2*, *IGFBP7*, and *IGFBP4* are among the most highly expressed DEGs, with the average log fold changes of 2.04–3.73 (Figs. 1c, 3a, 3b, and Supplementary Data 1). These insulin-like growth factor regulating proteins[32] play crucial roles in tumorigenesis via modulation of cancer hallmarks and involve in intracellular tumorigenic signal transduction[33]. *TIMP1* (tissue inhibitor of matrix

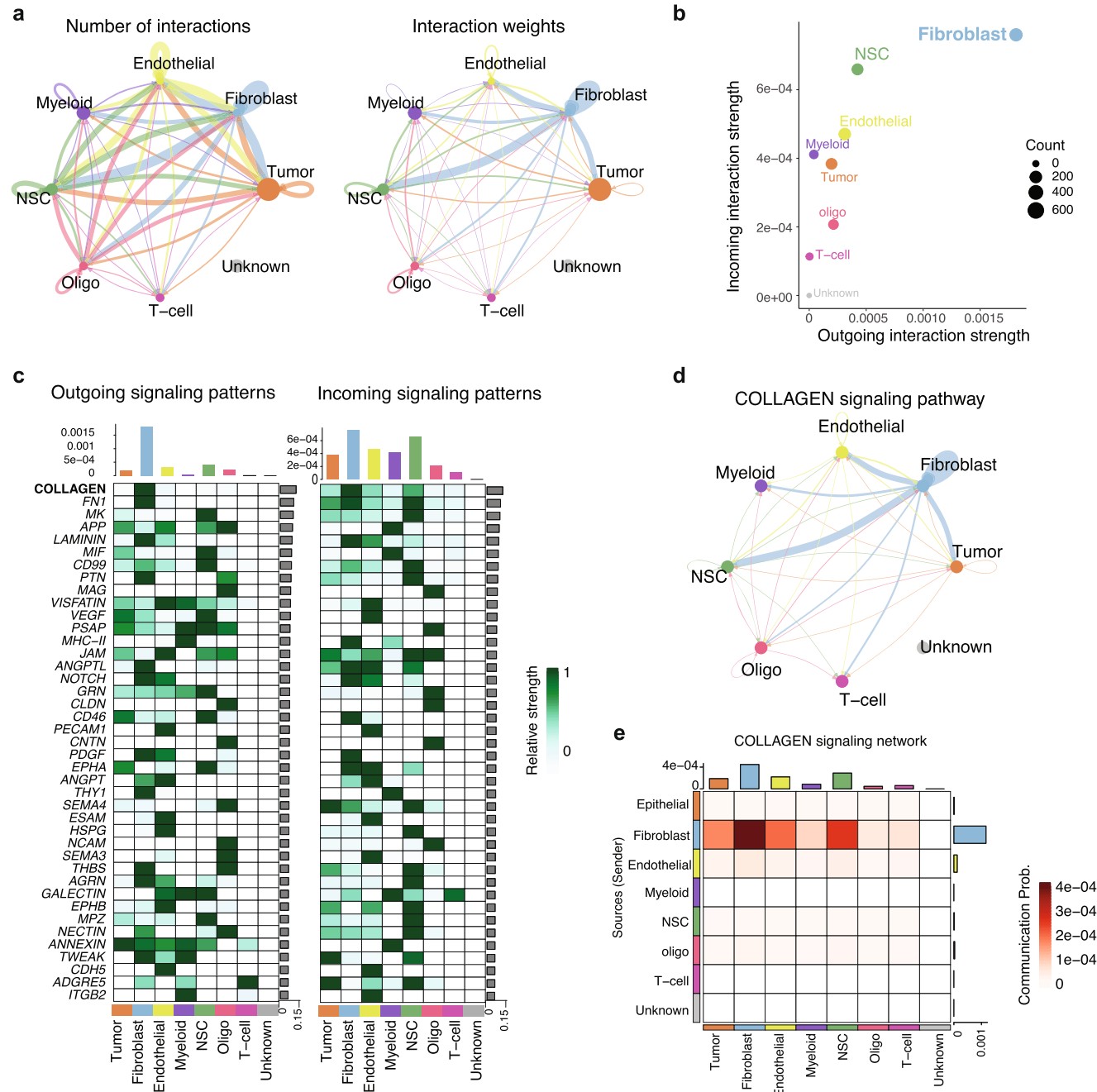

**Fig. 2 Inference of cell–cell communications in TME. a** Cell–cell communications between the identified cell types. **b** Illustration of the incoming and outgoing interaction strengths for each of the cell types. **c** The incoming and outgoing signaling pathways of each cell type. **d** The cell–cell interaction network among different cell types regarding the collagen signaling pathway. **e** The heatmap shows the communication probability of the collagen signaling pathway.

metalloproteinase 1), a well-known prognostic predictor of poor survival and hallmark gene for tumor-associated fibroblast[34], is also among the top DEGs (LFC: 3.55). Accordingly, *MMP9* and *LOXL2*, which are often co-expressed with *TIMP1* during breast tumor metastasis[35], present overexpression in fibroblasts (LFC: 1.98, and 1.02, respectively). Intriguingly, *TIMP3*, which is often considered as a protective factor in tumors, is shown with over-expression in fibroblasts (LFC: 2.49). *SPARC* is also highly expressed in fibroblasts (LFC: 2.76), which has been associated with bone-related breast and lung cancer invasion and metastasis[36,37]. Another interesting observation is that, the oncogene platelet-derived growth factor receptor β (*PDGFRB*)

related with angiogenesis in tumor tissues is upregulated in fibroblasts (LFC: 2.02), rather than the brain fibroblast hallmark gene *PDGFRA* (LFC: 0.25)[38,39]. *OLFML3*, *TMEM119*, and *ABI3*, which are related with the activation of microglia or macrophage as well as angiogenesis[40–42], are present in both myeloid cells and fibroblasts.

With the abundance of the hallmark genes as well as tumor-related genes in fibroblasts, gene set enrichment analysis (Fig. 3c) identifies that the up-regulated genes in fibroblasts are associated with ECM organization and collagen formation pathways. In contrast, the down-regulated genes are associated with the creation of complement C4 and C2 activators pathway and

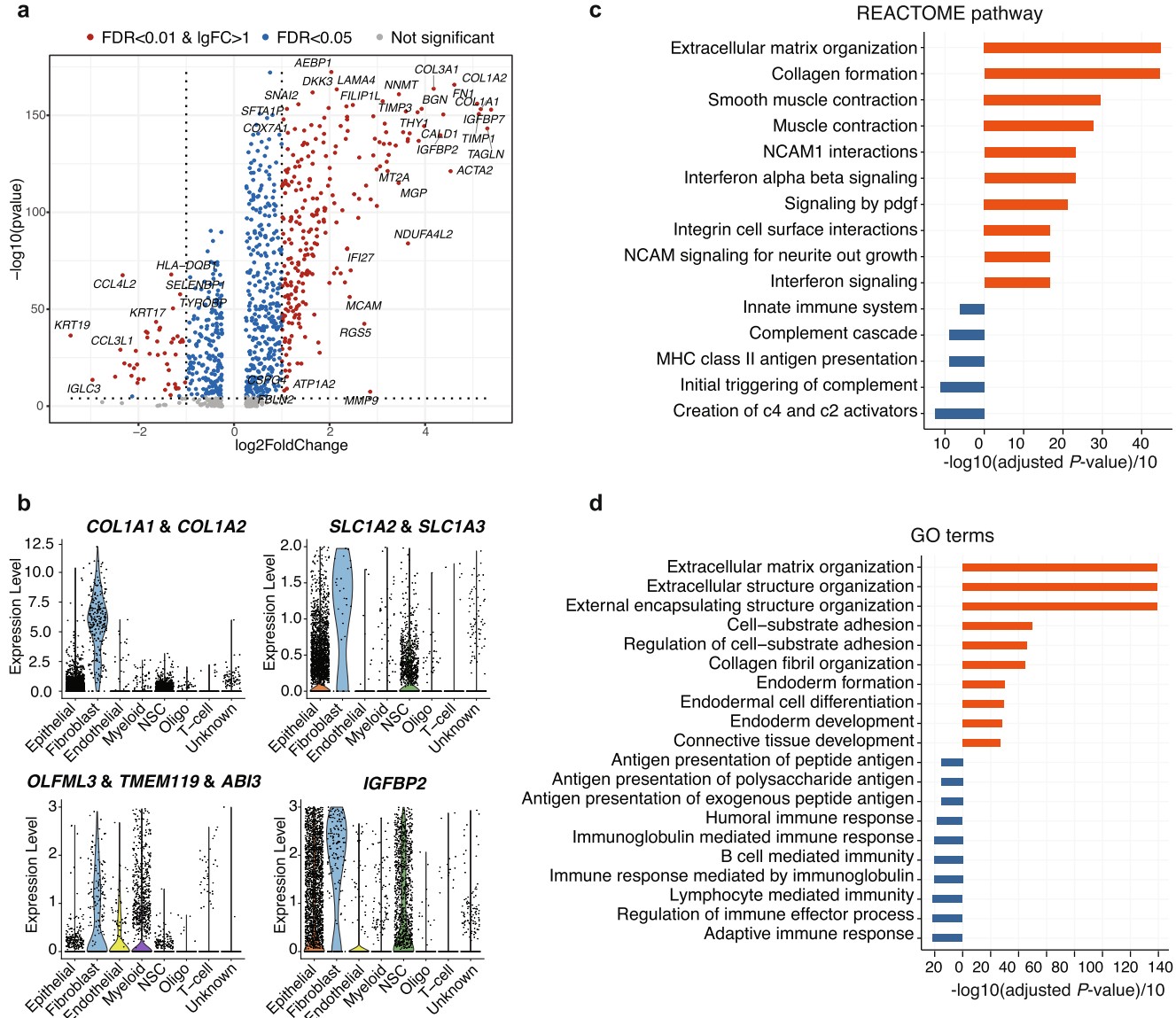

**Fig. 3 Characterization of tumor-associated fibroblasts. a** The volcano plot shows the differentially expressed genes (DEGs) between fibroblasts versus the rest of cells. The *x*-axis represents the log2 (fold change) of the DEGs and the *y*-axis represents the adjusted *P*-value (−1 × log10 scale). Blue dots represent genes with adjusted *P*-value < 0.05. Red dots represent the genes with adjusted *P*-value < 0.01 and |log2 FC|>1. **b** Violin plots show the expressions of functional genes that are abundant in fibroblasts. **c** Significantly enriched REACTOME pathways of the DEGs of fibroblasts are shown in bar plots. The *x*-axis represents -log10(adj. *P*)/10, which is calculated by the gene set enrichment test. **d** Significantly enriched GO terms of the DEGs of fibroblasts are shown in bar plots. The *x*-axis represents -log10(adj. *P*)/10, which is calculated by the gene set enrichment test.

the complement cascade pathway, which may be related with the inactive complement-related immune microenvironment in the brain metastatic tissues[43]. Further enrichment based on GO (Gene Ontology) terms consistently reveal the ECM organization and extracellular structure organization (Fig. 3d). Taken together, these results indicate the specific functions of tumor-associated fibroblasts, with altered gene expression patterns different with healthy fibroblasts[38], suggesting their potential impacts on brain metastasis.

**Fibroblast-centered cell–cell interactions**. As the observed tumor-associated fibroblasts play a central role in the intercellular communications at the brain metastatic tumor tissues, we further explore the ligand-receptor pairs involved in the major cellular crosstalk between fibroblasts and three other cell populations, including tumor cells, myeloid cells, and NSC-like stroma, which

show the strongest communications with fibroblasts (Fig. 4a). Noteworthy, strong communication probabilities mostly occur between ligands including collagens (type I, IV, and VI) and fibronectin (*FN1*), with receptors including integrin α_Vβ_8 (*ITGAV* and *ITGB8*), syndecan 1 and 4 (*SDC1* and *SDC4*), and *CD44*. These ligands and receptors dominate the active signaling between fibroblasts and tumor cells (Fig. 4b). Meanwhile, the corresponding receptors (integrin α_Vβ_8, syndecan 1 and 4, and *CD44*) are actively expressed in tumor cells (Fig. 4c). The major communications between myeloid cells and fibroblasts are *CD44*-based. The NSC-like stromal cells demonstrate similar communication patterns with fibroblast as the tumor cells. Among all ligands secreted by fibroblasts, collagen type I and fibronectin are important across the interactions with all the three cell types. Meanwhile, *CD44* is the common receptor shared by all the three cell types. *CD44* is known to be associated with TME[44], but its role as a common receptor in the brain metastatic tissue is first documented. Taken together,

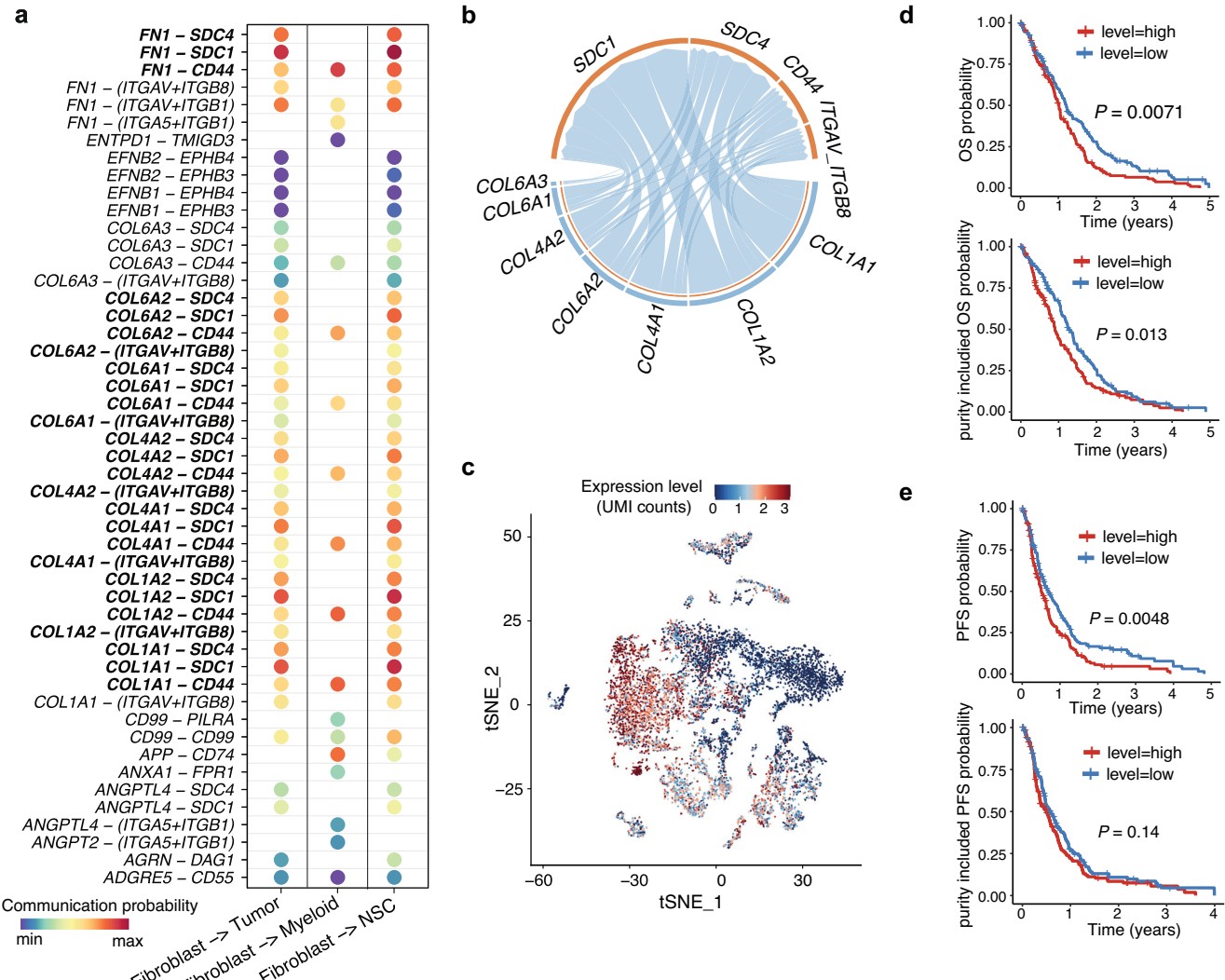

**Fig. 4 Significant tumor receptors for patient's prognosis. a** The significantly related ligand-receptor interactions of mostly communicated cell types. **b** Highly communicated ligand-receptor interactions between fibroblasts and tumor cells. **c** t-SNE plot shows the expressions of active tumor receptors including *ITGAV*, *ITGB8*, *SDC1*, *SDC4*, and *CD44*. **d** The upper panel shows the Kaplan-Meier (KM) overall survival curves for TCGA GBM patients, which are stratified by the median expression of the active tumor receptors. The lower panel shows the KM overall survival curves for GBM patients, which are stratified by the median expression of the active tumor receptors considering tumor purity. The *y*-axis represents the probability of overall survival, and the *x*-axis is time in days. **e** The upper panel shows the Kaplan-Meier (KM) progression-free survival (PFS) curves for GBM patients, which are stratified by the median expression of the active tumor receptors. The lower panel shows the KM PFS curves for GBM patients, which are stratified by the median expression of the active tumor receptors considering tumor purity. The *y*-axis represents the probability of PFS, and the *x*-axis is time in days.

these results suggest that the identified tumor-associated fibroblasts contribute to a unique ECM which is featured by collagen type I and fibronectin, and mediate the activities of tumor, myeloid cells, and NSC-like stromal cells.

**Significant tumor receptors for patient's prognosis.** Next, we evaluate whether the high expressions of collagen type I receptors (*ITGAV*, *ITGB8*, *SDC1*, *SDC4*, and *CD44*) in tumor tissues, i.e., tumor receptors, are related with the prognosis of patients with brain tumor. Since we have demonstrated this in patients with brain metastasis in our previous work[10], herein we further examine whether this remains true in glioblastoma patients. Remarkably, significant associations are observed between the increased expressions of tumor receptors and decreased overall survival of GBM patients from TCGA (Fig. 4d). The statistical significance of the difference between the K-M survival curves for patients in the high-risk group (red) and low-risk group (blue) is

assessed using the log-rank test, with *P*-values of 0.007. Since the TCGA is bulk-seq data, we consider the tumor purity of these patients and still identify a significant association between tumor receptors with patient overall survival (*P*-value = 0.013). Such significant association is also observed in terms of progression free survival (PFS; Fig. 4e). Collectively, the tumor receptors, i.e., the collagen type I receptors, possess convincingly strong prognostic power in brain tumor patients, even with the tumor purity considered.

**Interrogation of tumor-associated fibroblasts in external single-cell RNA-seq data.** From our in-house data, we reveal that type I collagen secreting tumor-associated fibroblasts is a key mediator in metastatic brain tumors, with the involved tumor receptors associating with patients' survival. To further validate our findings in other publicly available datasets, we include a recently published single-cell RNA-seq data from brain metastasis patient samples[26],

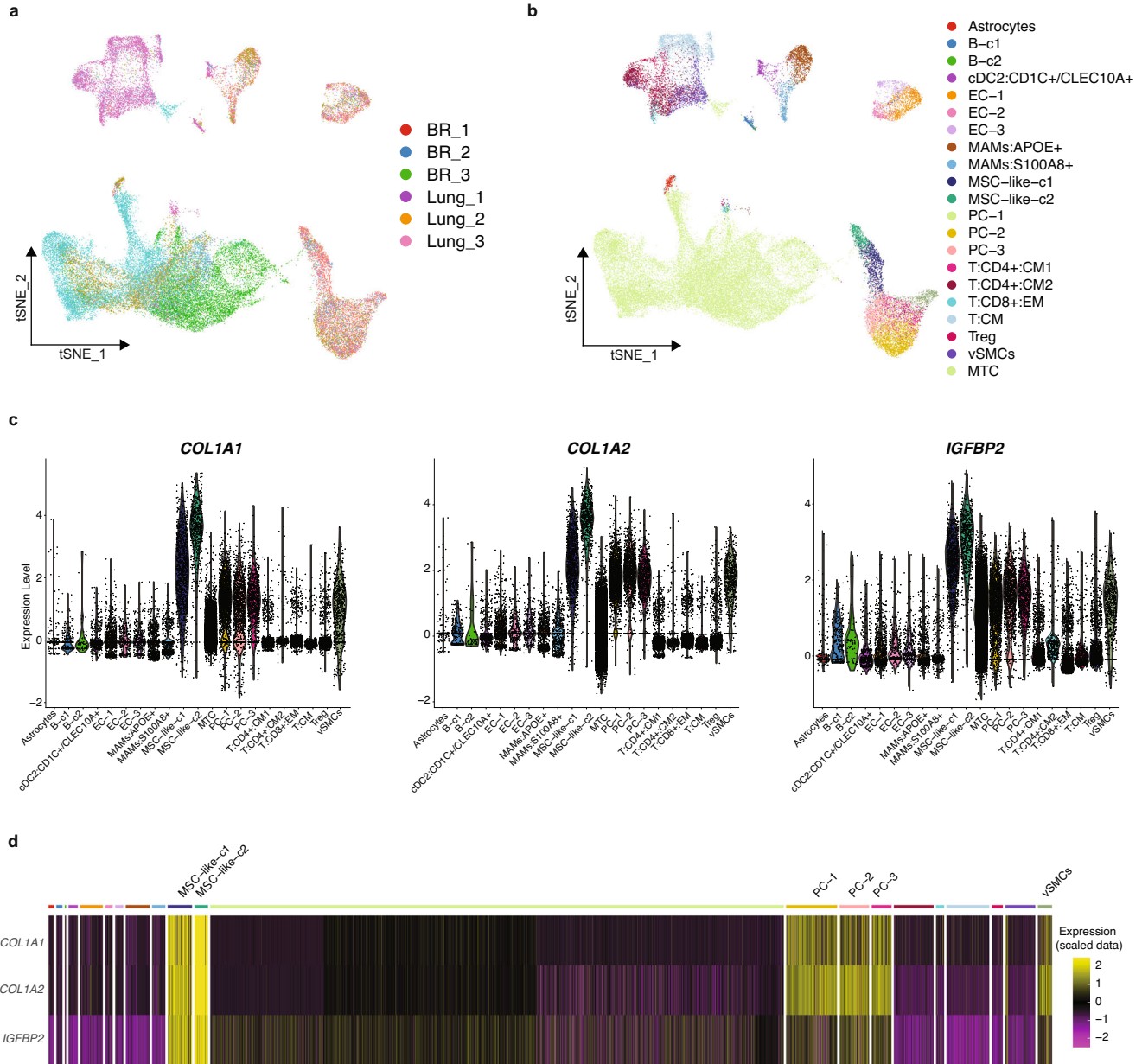

**Fig. 5 Identification of tumor-associated fibroblasts in external scRNA-seq data. a** t-SNE plot of the profiled single cells. Cells with the same colors are from the same patient sample. **b** t-SNE plot of the identified cell identities. **c** Violin plots show the expressions of functional genes that are abundant in fibroblasts. **d** Heatmap shows the expressions of fibroblast signatures in each cell type.

which consist of 3 breast cancer brain metastasis and 3 lung cancer brain metastasis samples (Fig. 5a). Cell type annotations provided by Gonzalez et al.[26] are shown in Fig. 5b. As shown in Fig. 5c, mesenchymal stromal cell-like 2 (MSC-like-c2) cells uniformly and highly express the fibroblast signatures (*COL1A1*, *COL1A2*, and *IGFBP2*). Other cell populations including mesenchymal stromal cell-like 1 (MSC-like-c1), as well as mural vascular cells (MVC) including pericytes (PC), and vascular smooth muscle cells (vSMCs) (PC-1, PC-2, PC-3, and vSMCs), majority of cells also show strong expressions of these signatures. Figure 5d further demonstrates that the fibroblast-like cell populations (PC-1, PC-2, PC-3, MSC-like-c1, MSC-like-c2, vSMCs) especially MSC-like-c2 have similar molecular property with the tumor-associated fibroblasts identified in our in-house data.

Based on this external dataset[26], we then use the CellChat[27] tool to infer the numbers and the strength of interactions among different cell populations in brain metastasis samples (Fig. 6a).

Consistent with our results (Fig. 2a), most interactions are observed among fibroblast-like cells (mainly MSC-like-c2, also including MSC-like-c1, PC-1, PC-2, PC-3, and vSMCs) and endothelial cells (EC-1, EC-2, EC-3). When considering the interaction strength (represented by the interaction weights in Fig. 6b), fibroblast-like cells especially MSC-like-c2 demonstrate a pivot role in the tumor microenvironment. These secondary analysis results of external scRNA-seq data further confirm that these *COL1A1+/COL1A2+/IGFBP2+* fibroblast-like cells are tumor associated and play a central role in the TME of brain metastasis tissues.

As the observed fibroblast-like cells played a central role in the intercellular communications at the brain metastatic tumor tissues, we further explore the interactions between other cell types with tumor cells in the external data (Fig. 6c). Notably, as expected, the *COL1A1+/COL1A2+/IGFBP2+* fibroblast-like cells especially MSC-like-c2 present strong interactions with malignant

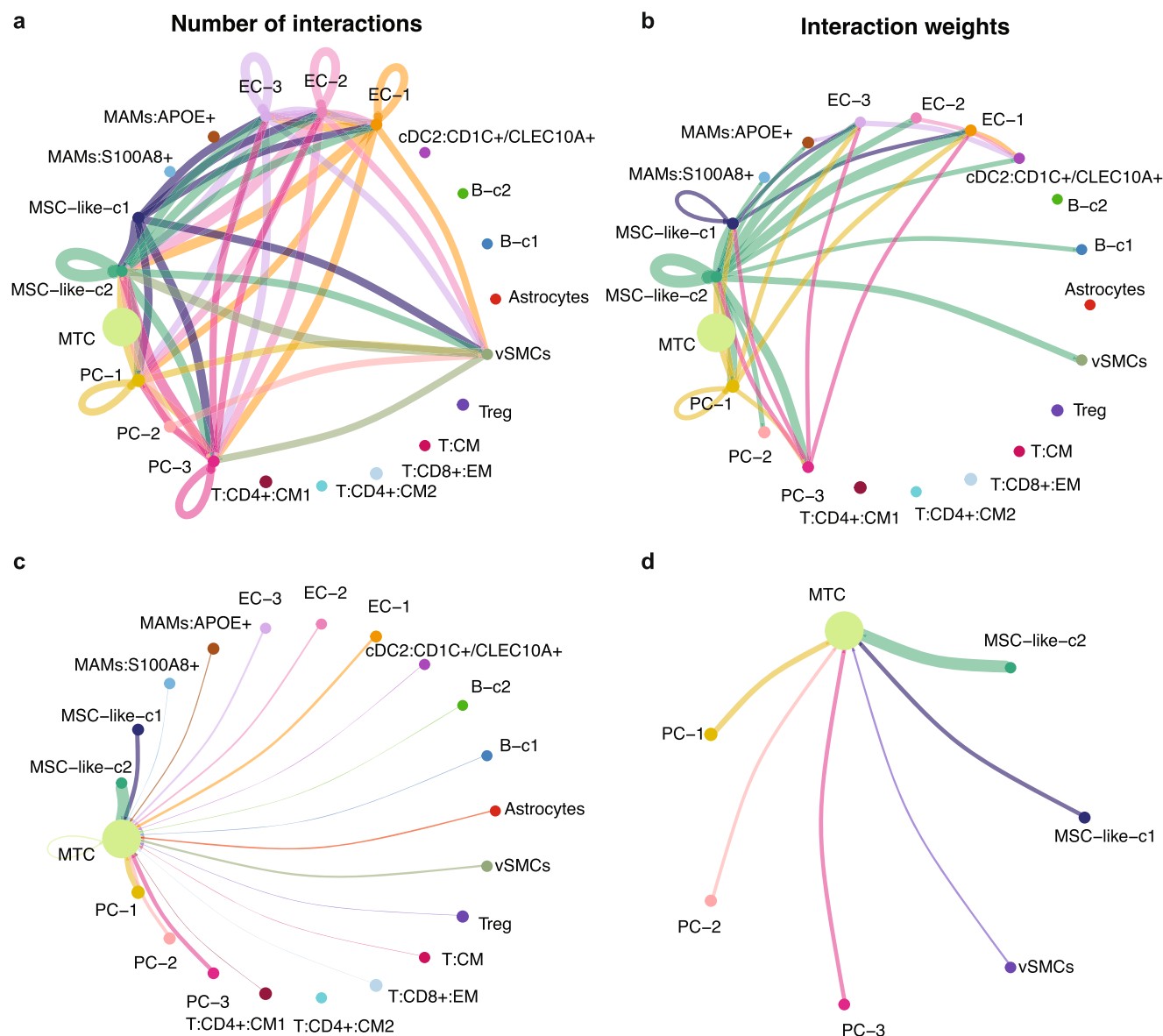

**Fig. 6 Cell–cell communications in external scRNA-seq data. a** Cell–cell communications between different cell types. **b** Illustration of the incoming and outgoing interaction strengths between different cell types. **c** Cell–cell communications between the identified cell types with tumor cells. **d** Cell–cell communications between the mostly communicated cell types with tumor cells.

tumor cells (MTC), which confirms our observation from in-house data. The intercellular communications between only fibroblast-like cells and tumor cells are shown in Fig. 6d. Examination of the interactions between fibroblast-like cells and tumor cells (Fig. 7) shows that strong communications occur between ligands including collagens (type I, IV, and VI) and fibronectin (*FN1*), with receptors (*SDC1*, *SDC4*, and *CD44*), which indicates that the tumor-associated fibroblasts potentially contribute to a unique ECM and mediate the activities of tumor cells.

## Discussion

Brain metastases are the most common intracranial tumor and is prevalent among most cancer cases (~20%)[1]. It is a devastating complication of systemic disease with limited treatment options and portends a poor survival. Despite intensive efforts to characterize biomarkers, there remains a limited improvement in metastatic brain prognosis during the past decades. Patients with

brain metastasis are often diagnosed with advanced systemic disease burden, leading to a 5-year overall survival rate below 10%[45]. Though there are unresolved issues regarding brain metastasis management and surveillance, attempts have been made to predict outcomes that would help to triage patients to the proper monitoring and treatment modalities. For example, studies have indicated that patients who are at high risk of experiencing rapid and numerous distant brain failures may serve as better candidates for whole-brain radiotherapy versus radiosurgery[46–48]. Other symptoms such as the development of leptomeningeal disease[49] and radiation necrosis[50] would benefit from additional predictive biomarkers. Therefore, the identification of effective biomarkers to promote efficient surveillance and drug guidance in brain metastasis is in critical need. In this work, we applied single-cell RNA-seq and focused on investigating the cell-cell interactions. We identified the pivotal role of tumor-associated fibroblasts in the brain metastatic tissues as well as the collagen/fibronectin-CD44/integrin $\alpha_V\beta_8$/syndecans ligand-receptor axis which serve as the molecular underpins in the

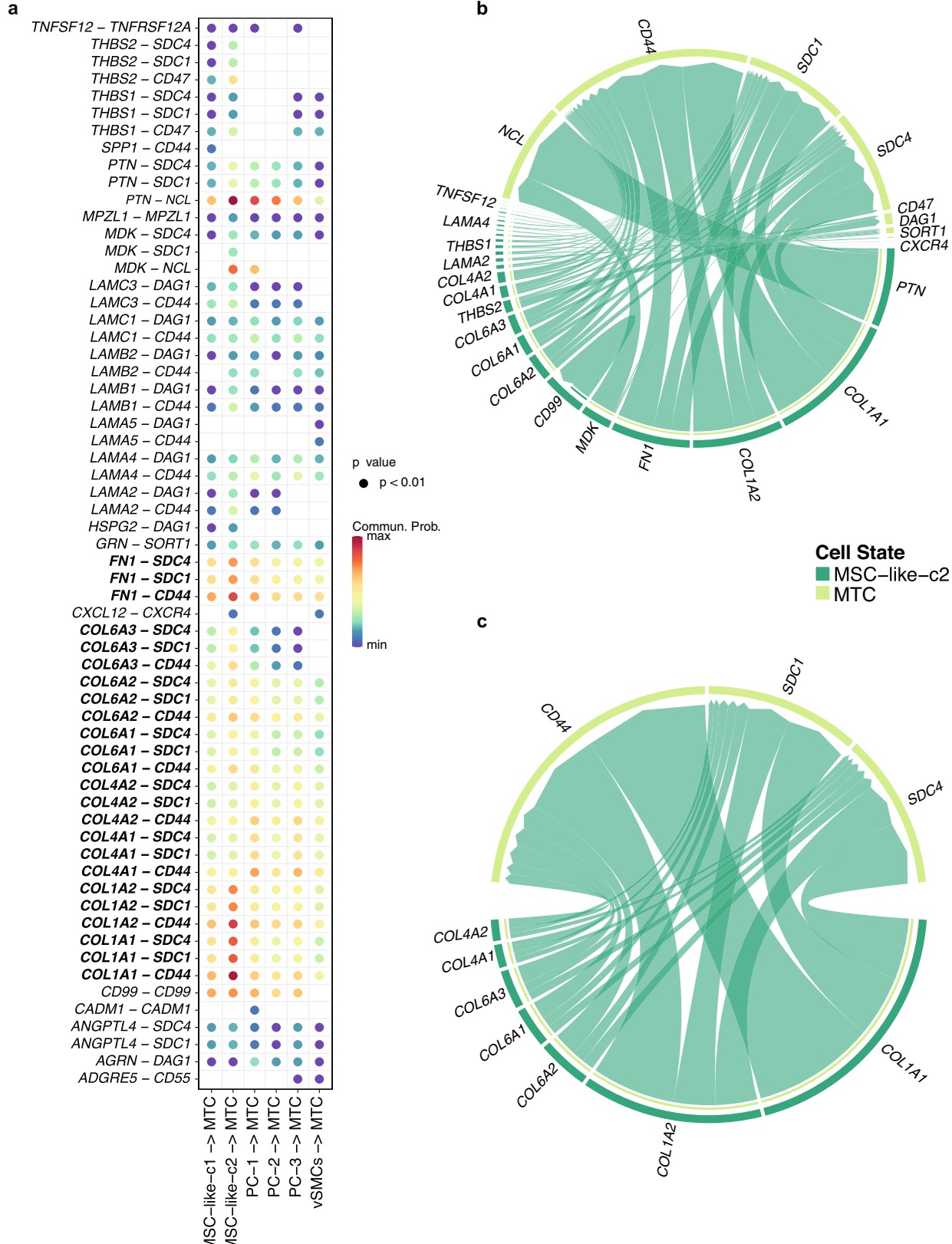

**Fig. 7 Ligand-receptor interactions identified in external scRNA-seq data. a** The significantly related ligand-receptor interactions of mostly communicated cell types with tumor cells. **b** Highly communicated ligand-receptor interactions between MSC−like−c2 cells and tumor cells. **c** Highly communicated collagen-based ligand-receptor interactions between MSC−like−c2 cells and tumor cells. MSC-like-c2 refers to mesenchymal stromal cell-like-c2 cells.

TME. Future studies are warranted to thoroughly characterize in detail each of the immune cell types present in the tumor lesions.

We have recently reported that, according to the bulk multi-omics data generated from the brain metastatic tissues, collagen type I (COL1A1 and COL1A2) is a predictive protein biomarker for higher risk of leptomeningeal disease after craniotomy and radiation[10]. Our discoveries from the single-cell RNA-seq data partially reveal the underlying molecular mechanisms that explain the collagen gene as the predictive biomarker in brain metastasis. Moreover, as therapeutic effects can be mediated by multifaceted interactions between different cell types[51], the collagen signaling events behind fibroblast-tumor communications may facilitate improving treatment effectiveness and developing successful therapies.

The central role of tumor-associated fibroblasts in the brain TME may be related with wound healing mechanism, since cancer metastasis can be viewed as a chronically developed wound that is never healed[52]. The ECM in brain TME includes collagens, proteoglycans, and glycoproteins[28,53] that provide both physical and chemical cues that affect cancer progression and metastasis[54]. The activation of wound-healing mechanisms in tumor-associated fibroblasts as well as the M1-activated microglial and macrophage cells remodel ECM and favor tumor growth. Type I collagen is the major ECM component in lung and breast tissues where the primary tumor cells are origin from but are not abundant in brain tissues. The M1 activation of the myeloid cells suggests that the chronic wound healing mechanism might be hijacked by tumor cells, which activates the high expression of collagens, especially collagen type I, in fibroblasts and causes fibrosis and glial scar formation[22,55]. Such remodeled brain TME mimics the original TME in lung and breast tissues. Tumor cells demonstrate highly expressed type I collagen binding proteins, suggesting that tumor cells actively respond to such type I collagen rich ECM.

Moreover, our cell-cell interaction results of the signaling between collagen genes and tumor receptors including SDC1, SDC4, and CD44 are also indicated in relevant studies. Chute et al. report that the SDC1 is aberrantly induced in breast carcinomas, which stimulates tumor cell growth and orchestrates stromal extracellular matrix fiber alignment, thus fostering the microenvironment for breast cancer brain metastasis[56]. Meanwhile, the collagen-CD44 interaction is shown to contribute to collagen accumulation or organization in a context-dependent way[57]. Type I collagen is not abundant in normal brain tissues, and the functions of brain cells do not depend on the type I collagen related cell adhesion. Hence, the interactions between tumor associated fibroblasts and tumor cells as well as the observed M1 activation of native microglial cells and infiltrated macrophages serve as vital players in brain metastasis and may contribute to the discovery of promising therapeutic targets for treatment.

## Methods

**Human sample acquisition.** Patients were enrolled in an IRB-approved prospective clinical trial at Wake Forest School of Medicine. Patients who were eligible for this study had a history of either lung cancer or breast cancer and had a suspected brain metastasis on imaging that would require surgical resection followed by post-operative cavity-directed radiosurgery. Patients were considered ineligible if they had the pre-existing leptomeningeal disease, >10 brain metastases, or performance status <60. Patients were enrolled in the study after giving informed consent that would allow a portion of their surgical resection specimen to be processed for single-cell sequencing. In this study, samples from 5 patients diagnosed with brain metastasis of breast (n = 3) and lung (n = 2) origins were included. Fresh remnant tumor tissues were collected at the time of elective curative resection by the Tumor Tissue and Pathology Shared Resource (TTPSR) of the Wake Forest Baptist Medical Center Comprehensive Cancer Center (WFBMC-CCC). Acquisition of de-identified samples from the TTPSR for single-cell isolation and research use was in accordance with approved Institutional Review Board

protocol. All relevant ethical regulations were followed and informed consents were obtained from participants.

**Patient treatment and follow-up.** Cavity-directed radiosurgery was performed on the Gamma Knife Perfexion. Treatments were generally performed within 6 weeks of surgical resection. A 3 Tesla MRI was performed on the day of radiosurgery. Evaluation of this MRI was performed by faculty radiologist on the day of radiosurgery in order to optimally detect any occult metastases. All metastases found on the day of radiosurgery were treated according to the dose recommendations from RTOG 90-05. Patients were generally followed clinically and with an MRI of the brain 4–8 weeks after radiosurgery and then every 3 months for the first two years. If no evidence of tumor progression or recurrence, MRI's were performed less frequently thereafter.

**Tissue dissociation and single-cell RNA sequencing.** Fresh tissue samples were obtained by the TTPSR and placed in a specific tissue storage medium (Miltenyi) before being stored at a temperature of 4 °C. Within a 24-hour timeframe, the tissues underwent a series of procedures to obtain single-cell suspensions. This involved utilizing the human tumor dissociation kit and GentleMACS protocols, followed by the removal of red blood cells (RBCs) through negative selection using CD235a microbeads (Miltenyi) based on recommended methods. The number of recovered cells was determined by performing trypan blue exclusion using an automated counter known as LUNA II. To ensure long-term viability, the cells were frozen while maintaining their viability in a solution consisting of 10% Hybri-Max DMSO (Sigma-Aldrich) and 90% heat-inactivated fetal bovine serum (FBS). This was achieved by gradually cooling them in isopropanol at a rate of −1 °C per minute and then storing them under liquid nitrogen vapor. In preparation for scRNA-seq, the cells were thawed and subjected to a washing procedure following the established protocol provided by 10x Genomics.

The Cancer Genomics Shared Resource (CGSR) at the WFBMC-CCC conducted all scRNA-seq procedures. Suspensions containing viable cells, with an average viability of 83.4 ± 9.9% (n = 5), and cell concentration averaging 995 ± 267 cell/μl, were loaded into wells on a 10X Chromium single-cell capture chip. The aim was to achieve a cell recovery rate of 2000–3000 cells. Using the Chromium Single Cell Controller, single-cell gel beads in emulsion (GEMs) were generated. Subsequently, scRNA-seq libraries were prepared using the Chromium Single Cell 3' Library and Gel Bead kit, following the manufacturer's protocol provided by 10x Genomics. For sequencing, the prepared libraries were loaded onto an Illumina NextSeq500 instrument with a High Output 150 cycle kit. Paired-end sequencing was performed with the following read lengths: 26 bp for Read1, 8 bp for i7 Index, 0 bp for i5 Index, and 98 bp for Read2. This allowed for comprehensive sequencing of the scRNA-seq libraries, enabling further analysis and investigation of the captured single-cell transcriptomes.

**Single-cell RNA sequencing data processing.** To perform sample de-multiplexing, alignment, filtering, and UMI (universal molecular identifier) counting, the Cell Ranger Single Cell Software Suite v.2.0.1 was employed (source: https://support.10xgenomics.com/single-cell-gene-expression/software/pipelines/latest/what-is-cell-ranger). The transcriptomic data of each specific subpopulation were aggregated to enable direct comparisons between single-cell transcriptomes. Cells that exhibited low quality were excluded from the analysis if the number of expressed genes fell below 200. Additionally, cells were eliminated if their proportions of mitochondrial gene expression exceeded 40%. Following the quality control measures, a total of 10,896 single cells from 5 samples were successfully captured. The number of cells recovered per channel ranged from 640 to 2753, showcasing variability across the samples. The mean reads with Unique Molecular Indexes (UMIs) per cell ranged from 51,097 to 211,783, indicating variation in the depth of sequencing for individual cells.

In the subsequent analysis, we utilized the Seurat toolkit[58] for dimension reduction and cell clustering. The data was initially loaded into R as a count matrix and log-transformed using the "NormalizeData" function. To address batch effects, we integrated the count matrix from two samples using the "IntegrateData" function, resulting in a batch-corrected expression matrix. Subsequently, Principal Component Analysis (PCA) was performed on this batch-corrected data, focusing on the first 30 principal components. For cell clustering and visualization using t-distributed stochastic neighbor embedding (t-SNE), we employed the first 30 principal components derived from PCA. To determine the optimal number of cell clusters, we calculated silhouette scores for various cluster numbers, ranging from five to twenty clusters. These scores measure the similarity of cells within a cluster compared to cells in other clusters. The optimal number of clusters was determined by maximizing the Silhouette score. Additionally, we attempted to employ the RaceID[59] tool for identifying rare cells. However, in our single-cell data, we did not detect any significant rare cell populations. The cell annotations were then assigned based on the expressions of corresponding marker genes and further confirmed by collaborating biologists who are co-authors of this study.

**Statistics and Reproducibility.** Reactome pathways[60] and GO terms were downloaded from the MSigDB Collections[61]. Functional enrichment was assessed by a hypergeometric test, which was used to identify a priori-defined gene set that showed statistically significant differences between two given clusters. The gene set

enrichment analysis was performed by the clusterProfiler package[62]. Test P values were further adjusted by Benjamini-Hochberg correction, and adjusted P values less than 0.05 were considered statistically significant. Differentially expressed genes (DEGs) were identified using the "FindMarkers" function in the R package of "Seurat"[63]. DEGs were evaluated with the Bonferroni-adjusted P-value (adj. $P < 0.05$) and the absolute log2-fold-change (|log2 FC| >= 1). All statistical analyses are performed using R v.4.2.0.

**Reporting summary**. Further information on research design is available in the Nature Portfolio Reporting Summary linked to this article.

## Data availability

Our generated single-cell RNA-seq data is available in the NCBI Gene expression Omnibus database (GEO) with accession number GSE234832. External single-cell data used in this study is available in GEO with accession number GSE186344. Source data underlying figures are provided in Supplementary Data 2.

## Code availability

The analysis codes for survival analysis are shared in GitHub and Zenodo (https://github.com/QSong-github/BrMet)[64].

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

## Acknowledgements

The work is partially supported by the Cancer Center Support Grant from the National Cancer Institute to the Comprehensive Cancer Center of Wake Forest Baptist Medical Center (P30 CA012197), and the Lung Cancer Initiative Foundation Grant. The content is solely the responsibility of the authors and does not necessarily represent the official views of the National Institutes of Health. Q.S. is supported in part by the Bioinformatics Shared Resources under the NCI Cancer Center Support Grant to the Comprehensive Cancer Center of Wake Forest University Health Sciences (P30CA012197). J.S. is partially financially supported by the Indiana University Precision Health Initiative and by the Indiana University Melvin and Bren Simon Comprehensive Cancer Center Support Grant from the National Cancer Institute (P30 CA 082709).

## Author contributions

J.S., J.R., and M.D.C. conceived the study design. M.H.S and S.S.O. led the patient recruitment and biospecimen collection. M.H.S., L.C., and L.D.M. generated the single-cell RNA-Seq data. S.S.O., J.R., M.D.C., and A.K.P. collected and organized the clinical data. Q.S. and J.S. designed and performed bioinformatics analysis. F.X., L.D.M., H.L., and K.W. reviewed and contribute to the biological components. Q.S. and J.S. led the manuscript preparation with the contributions of all authors. All authors have approved the final version of the manuscript.

## Competing interests

The authors declare no competing interests.

## Ethics approval and consent to participate

All tissues and clinical data utilized in this study were obtained for research use by patient consent under an institutional review board-approved protocol at Wake Forest Baptist Medical Center, Winston Salem, NC.
