## [Peer Review File · Communications Biology]

Reviewers' comments:

Reviewer #1 (Remarks to the Author):

The authors generated scRNA seq dataset for 5 metastatic tumors in the brain. The key finding of this article was that fibroblasts played a central role in remodeling tumor microenvironments through the type I collagen signaling axis, providing a potential biomarker and target. They also showed that this process was not only found in patients with metastatic brain cancer but also in glioblastoma. Though lacks experimental validations, the paper is well-organized and logically developed. Here are a few specific comments,

Major Comments

Characteristics of fibroblasts are obtained by comparing fibroblasts to other cell types, rather than to fibroblasts under normal conditions, which may result in general features of fibroblasts rather than features specific to tumor-associated fibroblasts.

Figure 2E shows that the cell cluster "Endo1" is the major source of collagen signaling in the cell-cell communication network of the TME. This seems to contradict the results shown in figure 2D.

From lines 141 to 145 the authors state that various changes in the proportion of cell types were observed among different tumors. However, it is difficult to draw these conclusions from such a small sample size.

A few slides of immunofluorescence staining showing the colocalization of the reported ligand and receptors on the corresponding cell types in the brain cancer TME would greatly strengthen the paper's claim. Or any direct experimental validation of such interactions in the TME would be great.

Minor Comments

I would suggest the authors use 5 years survival and PFS analysis in Figures 4D and 4E instead of 4000 days, as the survival and PFS of patients for a longer period may be confounded by many other factors, such as gender and age, etc.

P value or any statistic measure of figure 3C and 3D is missing.

Line:114, Figure 1A: In the main text, the author stated that the figures were UMAP projections, but tSNE was shown in the Figures. The same error appeared in the method section.

Line:167, Figure 2E: Endo1 and Endo2 were not previously defined.

Reviewer #2 (Remarks to the Author):

In this manuscript from Song et al., the authors have performed scRNA profiling of brain metastatic samples from five patients. The authors have used standard bioinformatics analysis and reported that the type I collagen-secreting tumor-associated fibroblasts as a key mediator in metastatic brain tumors and reveal the involved tumor receptors associated with patients' survival.

Major Comments/Suggestions

1. Though the data generated in this study could be a valuable asset for the scientific community; however, the authors may try to put some more effort into the computational front. For instance, the present analysis does not bring any new major scientific insights. While I agree with the authors that there are presently limited sc-datasets of Brain Metastasis, however, I am still worried that authors must present some better use case of this worthy data (may develop some computational method to dig more into the dataset or at least do the integrative analysis with other publicly available

datasets?).

2. The authors mentioned that they have used canonical/standard markers for the cluster annotation, which are indeed a handful for distinct cell types, and may lead to bias. I recommend the author try some automated/unbiased/data-driven software for cluster identity and then compare it with the classical markers. Please refer to this review that summarizes most of the widely adopted methods: <https://www.sciencedirect.com/science/article/pii/S2001037021000192>

3. In Figure 1D, the authors have annotated a cluster with an unknown identity. In this case, I really recommend authors work on point 2 as discussed above.

4. In UMAPs, the cluster size largely varies with the clustering parameters, such as Perplexity. Have the authors used some standard method to define the optimal number of clusters on their expression matrix, or have they used the default parameters? In the case of the latter, I recommend authors carefully reexamine this since they might miss some rare/previously uncharacterized cell types.

5. I also recommend authors try some of the computational tools for rare cell identification using scRNA seq (FIRE? CellSIUS?), as this might bring further novel insights to their work.

6. Could authors detail the statistics used for the Survival analysis? Can they share their code on GitHub for further review? In Fig 4 panel E, both the populations seem rather overlap in the survival curve. Why authors have used Average for patient stratification? They might use the median expression as a cutoff, or there are other statistically robust approaches for this. Please refer to this tutorial: <https://bioconnector.github.io/workshops/r-survival.html>

Reviewers' Comments to Author:

REVIEWER COMMENTS

Reviewer #1 (Remarks to the Author):

The authors generated scRNA seq dataset for 5 metastatic tumors in the brain. The key finding of this article was that fibroblasts played a central role in remodeling tumor microenvironments through the type I collagen signaling axis, providing a potential biomarker and target. They also showed that this process was not only found in patients with metastatic brain cancer but also in glioblastoma. Though lacks experimental validations, the paper is well-organized and logically developed. Here are a few specific comments,

Major Comments

Characteristics of fibroblasts are obtained by comparing fibroblasts to other cell types, rather than to fibroblasts under normal conditions, which may result in general features of fibroblasts rather than features specific to tumor-associated fibroblasts.

Response: Thanks for the reviewer's comments. We do agree with the reviewer that comparing tumor-associated fibroblast with normal ones is the best option to delineate the biological properties of fibroblasts in tumor microenvironment. However, since in this study only patients with brain metastasis are recruited, we do not have tissues from healthy donors. Another challenge is that fibroblasts are rare in normal brain tissues. A large normal brain tissue is necessary for characterizing fibroblasts under normal condition. If such normal fibroblasts are available, then we can further verify the tumor-specific molecular features of the COL1A1⁺/COL1A2⁺/IGFBP2⁺ fibroblasts in the brain metastatic tissues. For now, our data accumulation is still not sufficient for such work.

Meanwhile, in our recently published work¹, the unusual presence of alpha-1 type I collagen is observed in brain metastatic tissues and is associated with the risk of leptomeningeal failure. This provides the pathological evidence at protein level that type I collagen-secreting fibroblasts are tumor-associated fibroblasts that exist in brain metastatic tissues.

Figure 2E shows that the cell cluster "Endo1" is the major source of collagen signaling in the cell-cell communication network of the TME. This seems to contradict the results shown in figure 2D.

Response: Thanks for the reviewer's comments. Sorry for the typo in the cell type labels in Fig. 2E. We have corrected it accordingly in our revision.

From lines 141 to 145 the authors state that various changes in the proportion of cell types were observed among different tumors. However, it is difficult to draw these conclusions from such a small sample size.

Response: Thanks for the reviewer's comments and we are sorry for the misunderstanding. Our conclusion is that the cellular composition proportion varies across patients. We do agree that it is difficult to draw conclusions about the differences among brain metastasis cases based on the

samples size from our in-house data. In the revised manuscript, we rephrased the related sentences to assure rigorous description. Details are shown below (copy revised part here).

“While tumor cells and myeloid cells are consistently identified across patient samples, their relative proportions vary from patient to patient. For example, the proportion of oligodendrocytes in a lung cancer brain metastasis sample (LUBMET1) was significantly higher than in other samples. Proportional variances are also observed for other cell types. Overall, we found a large degree of variation in the cellular composition among the five specimens. Such phenotypic variability has also been reported in multiple cancer types (23-25), including brain metastatic tissues (Gonzalez et.al²).”

A few slides of immunofluorescence staining showing the colocalization of the reported ligand and receptors on the corresponding cell types in the brain cancer TME would greatly strengthen the paper’s claim. Or any direct experimental validation of such interactions in the TME would be great.

Response: Thanks for the reviewer’s comments. Since the patient recruitment of this study has been finished and closed, it is currently challenging for our team to recruit new patients and collect brain metastatic tissues for the recommended colocalization analysis. To validate the cell-cell interactions identified in our study, we used a public single-cell RNA-seq data from brain metastasis patient samples (Gonzalez et.al²) as an external validation. This dataset consists of 3 breast cancer brain metastasis and 3 lung cancer brain metastasis samples (Figure 5A). The cell type annotations provided by Gonzalez et.al were shown in Figure 5B. Same with the analysis in our study, we used the CellChat³ tool, to identify the cell-cell communications and validate our findings.

We first examined whether the COL1A1⁺/COL1A2⁺/IGFBP2⁺ fibroblasts discovered in our work (Figure 1C) could be identified in Gonzalez’s data. As shown in Figure 5C and Figure 5D, the mesenchymal stromal cell-like 2 (MSC-like-2) cells uniformly and highly expressed COL1A1, COL1A2, and IGFBP2. In other cell populations including mesenchymal stromal cell-like 1 (MSC-like-1), as well as mural vascular cells (MVC) including pericytes (PC), and vascular smooth muscle cells (vSMCs) (PC-1, PC-2, PC-3, and vSMCs), majority of cells also showed increased expressions of these three biomarkers. Figure 5D further demonstrated those cell populations (fibroblast-like: PC-1, PC-2, PC-3, MSC-like-c1, MSC-like-c2, vSMCs) especially MSC-like-c2 had similar molecular property regarding these three hallmark genes with the COL1A1⁺/COL1A2⁺/IGFBP2⁺ fibroblasts in our study.

Figure 5

Based on this external public data (Gonzalez et.al²), we then used the CellChat³ tool to infer the numbers and the strength of interactions among different cell populations in brain metastasis samples (Figure 6A). Consistent with our results (Figure 2A), most interactions were observed among fibroblast-like cells (mainly MSC-like-c2, also including MSC-like-c1, PC-1, PC-2, PC-3, and vSMCs) and endothelial cells (EC-1, EC-2, EC-3). When considering the interaction strength (represented by the interaction weights in Figure 6B), fibroblast-like cells especially MSC-like-c2 demonstrated a pivot role in the tumor microenvironment, which was consistent with our findings (Figure 2A). These secondary analysis results of external scRNA-seq data further confirmed that these COL1A1⁺/COL1A2⁺/IGFBP2⁺ fibroblast-like cells were tumor associated and played a central role in the TME of brain metastasis tissues.

As the observed fibroblast-like cells played a central role in the intercellular communications at the brain metastatic tumor tissues, we further explored the interactions between other cell types with tumor cells in the external data (Figure 6C). Notably, as expected, the COL1A1⁺/COL1A2⁺/IGFBP2⁺ fibroblast-like cells especially MSC-like-c2 presented strong interactions with malignant tumor cells (MTC), which confirmed our observation from in-house data (Figure 2D). The intercellular communications between only fibroblast-like cells and tumor cells were shown in Figure 6D.

Figure 6

Moreover, we further explored the ligand-receptor pairs involved in the major cellular crosstalk between fibroblast-like cells and tumor cells (Figure 7A) using the same approach in our work (Figure 4A). Noteworthy, strong communications mostly occurred between ligands including collagens (type I, IV, and VI) and fibronectin (FN1), with receptors including syndecan 1 and 4 (SDC1 and SDC4), and CD44. These ligands and receptors dominated the active signaling between fibroblast-like cells and tumor cells (Figure 7A). Among all ligands secreted by fibroblasts,

collagen type I and fibronectin were significant across the interactions with all the three cell types. Meanwhile, SDC1, SDC4, and CD44 were the common receptors shared by all fibroblast-like cells. These observations were consistent with our previous findings (Figure 4A), which indicated that the tumor-associated COL1A1⁺/COL1A2⁺/IGFBP2⁺ fibroblasts potentially contributed to a unique ECM which is featured by collagen type I and fibronectin, and mediated the activities of tumor cells.

Meanwhile, in our recently published work¹, the unusual deposition of alpha-1 type I collagen is observed in brain metastatic tissues and is associated with the risk of leptomeningeal failure. This provides further pathological evidence at protein level that those tumor-associated fibroblasts secrete type I collagen, which play roles in brain metastasis microenvironment.

Figure 7

Minor Comments

I would suggest the authors use 5 years survival and PFS analysis in Figures 4D and 4E instead of 4000 days, as the survival and PFS of patients for a longer period may be confounded by many other factors, such as gender and age, etc.

Response: Thanks for the reviewer's comments. We modified Figures 4D and 4E with the survival analysis limited to 5 years. The updated figures are shown in our revised Figure 4D and Figure 4E.

P value or any statistic measure of figure 3C and 3D is missing.

Response: Thanks for the reviewer's comments. We have added the P-value in Figure 3C and Figure 3D.

Line:114, Figure 1A: In the main text, the author stated that the figures were UMAP projections, but tSNE was shown in the Figures. The same error appeared in the method section.

Response: Thanks for the comments. We have corrected the errors accordingly.

Line:167, Figure 2E: Endo1 and Endo2 were not previously defined.

Response: Thanks for the comments. We mis-labeled Endo1 and Endo2. We have corrected the errors accordingly.

Reviewer #2 (Remarks to the Author):

In this manuscript from Song et al., the authors have performed scRNA profiling of brain metastatic samples from five patients. The authors have used standard bioinformatics analysis and reported that the type I collagen-secreting tumor-associated fibroblasts as a key mediator in metastatic brain tumors and reveal the involved tumor receptors associated with patients' survival.

Major Comments/Suggestions

1. Though the data generated in this study could be a valuable asset for the scientific community; however, the authors may try to put some more effort into the computational front. For instance, the present analysis does not bring any new major scientific insights. While I agree with the authors that there are presently limited sc-datasets of Brain Metastasis, however, I am still worried that authors must present some better use case of this worthy data (may develop some computational method to dig more into the dataset or at least do the integrative analysis with other publicly available datasets?).

Response: Thanks for the reviewer's comments. The major goal of this work is to report the newly discovered type I collagen secreting tumor-associated fibroblasts as a key mediator in metastatic brain tumors, and reveal that the involved tumor receptors are associated with patients' survival. Standard bioinformatics workflows are used intentionally so that our results will be generalizable, thus can be included into the growing knowledgebase and repository of single-cell RNA-seq data. Our discoveries provide new biomarkers for effective therapeutic targets and intervention strategies. To further validate our findings in other publicly available datasets, we followed the reviewer's suggestion and included a recently published single-cell RNA-seq data from brain metastasis patient samples (Gonzalez et.al²), which consisted of 3 breast cancer brain metastasis and 3 lung cancer brain metastasis samples (Figure 5A). The cell type annotations provided by Gonzalez et.al were shown in Figure 5B.

We first examined whether the COL1A1⁺/COL1A2⁺/IGFBP2⁺ fibroblasts discovered in our work (Figure 1C) could be identified in Gonzalez's data. As shown in Figure 5C and Figure 5D, the mesenchymal stromal cell-like 2 (MSC-like-2) cells uniformly and highly expressed COL1A1, COL1A2, and IGFBP2. In other cell populations including mesenchymal stromal cell-like 1 (MSC-like-1), as well as mural vascular cells (MVC) including pericytes (PC), and vascular smooth muscle cells (vSMCs) (PC-1, PC-2, PC-3, and vSMCs), majority of cells also showed increased expressions of these three biomarkers. Figure 5D further demonstrated those cell populations (fibroblast-like: PC-1, PC-2, PC-3, MSC-like-c1, MSC-like-c2, vSMCs) especially MSC-like-c2 had similar molecular property regarding these three hallmark genes with the COL1A1⁺/COL1A2⁺/IGFBP2⁺ fibroblasts in our study.

Figure 5

Based on this external public data (Gonzalez et.al²), we then used the CellChat³ tool to infer the numbers and the strength of interactions among different cell populations in brain metastasis samples (Figure 6A). Consistent with our results (Figure 2A), most interactions were observed among fibroblast-like cells (mainly MSC-like-c2, also including MSC-like-c1, PC-1, PC-2, PC-3, and vSMCs) and endothelial cells (EC-1, EC-2, EC-3). When considering the interaction strength (represented by the interaction weights in Figure 6B), fibroblast-like cells especially MSC-like-c2 demonstrated a pivot role in the tumor microenvironment, which was consistent with our findings (Figure 2A). These secondary analysis results of external scRNA-seq data further confirmed that these COL1A1⁺/COL1A2⁺/IGFBP2⁺ fibroblast-like cells were tumor associated and played a central role in the TME of brain metastasis tissues.

As the observed fibroblast-like cells played a central role in the intercellular communications at the brain metastatic tumor tissues, we further explored the interactions between other cell types

with tumor cells in the external data (Figure 6C). Notably, as expected, the COL1A1⁺/COL1A2⁺/IGFBP2⁺ fibroblast-like cells especially MSC-like-c2 presented strong interactions with malignant tumor cells (MTC), which confirmed our observation from in-house data (Figure 2D). The intercellular communications between only fibroblast-like cells and tumor cells were shown in Figure 6D.

Figure 6

Moreover, we further explored the ligand-receptor pairs involved in the major cellular crosstalk between fibroblast-like cells and tumor cells (Figure 7A) using the same approach in our work (Figure 4A). Noteworthy, strong communications mostly occurred between ligands including collagens (type I, IV, and VI) and fibronectin (FN1), with receptors including syndecan 1 and 4 (SDC1 and SDC4), and CD44. These ligands and receptors dominated the active signaling between fibroblast-like cells and tumor cells (Figure 7A). Among all ligands secreted by fibroblasts, collagen type I and fibronectin were significant across the interactions with all the three cell types. Meanwhile, SDC1, SDC4, and CD44 were the common receptors shared by all fibroblast-like cells. These observations were consistent with our previous findings (Figure 4A), which indicated

that the tumor-associated COL1A1⁺/COL1A2⁺/IGFBP2⁺ fibroblasts potentially contributed to a unique ECM which is featured by collagen type I and fibronectin, and mediated the activities of tumor cells.

Meanwhile, in our recently published work¹, the unusual deposition of alpha-1 type I collagen is observed in brain metastatic tissues and is associated with the risk of leptomeningeal failure. This provides further pathological evidence at protein level that those tumor-associated fibroblasts secrete type I collagen, which play roles in brain metastasis microenvironment.

Figure 7

2. The authors mentioned that they have used canonical/standard markers for the cluster annotation, which are indeed a handful for distinct cell types, and may lead to bias. I recommend the author try some automated/unbiased/data-driven software for cluster identity and then compare it with the classical markers. Please refer to this review that summarizes most of the widely adopted methods: <https://www.sciencedirect.com/science/article/pii/S2001037021000192> [sciencedirect.com]

Response: Thanks for the reviewer’s comments. We followed the review by Pasquini et al⁴ and used the automatic methods, including the scCATCH⁵, scType⁶, and singleR⁷. However, these automated methods generated mutually inconsistent annotations, suggesting that such methods were not reliable to be used in our dataset. Detailed annotations were shown as below. Considering the inconsistency of different annotation tools, we demonstrate that the automated computation tools are not ready/sufficient. Expert-guided annotations by biologists (authors: Drs. Lance Miller and Fei Xing) are critically necessary for delivering reliable cell identities.

3. In Figure 1D, the authors have annotated a cluster with an unknown identity. In this case, I really recommend authors work on point 2 as discussed above.

Response: Thanks for the reviewer's comments. As suggested by the reviewer, we referred to the review⁴ and used the automatic annotation tools, including the scCATCH⁵, scType⁶, and singleR⁷. However, we found that different methods identified very different annotations. Thus the manual annotations by our biologist experts were more reliable. Moreover, for the cluster with unknown identity, we looked into its differentially expressed genes (DEGs), which only included IGLC3, IGLC2, IGHG1, IGHA1. These genes were ambiguous and could not be used for the identification of cell identity. For scientific rigor, we prefer to be conservative and keep these cells labeled as unknown.

4. In UMAPs, the cluster size largely varies with the clustering parameters, such as Perplexity. Have the authors used some standard method to define the optimal number of clusters on their expression matrix, or have they used the default parameters? In the case of the latter, I recommend authors carefully reexamine this since they might miss some rare/previously uncharacterized cell types.

Response: Thanks for the reviewer's comments. We used the t-SNE plots only for visualization, but not for cell clustering. Therefore, the parameters in t-SNE plot did not affect the clustering results.

We follow the standard workflow of single-cell clustering and annotation, which is widely used by other studies^{2,8-10}. As for t-SNE visualization, we also follow the standard workflow of Seurat¹¹. To determine the optimal numbers of cell clusters, we use the silhouette scores for different numbers of clusters, ranging from five clusters to twenty clusters, to measure how similar cells are in a cluster comparing with cells in other clusters. The optimum number of clusters is determined by maximizing the Silhouette score. Meanwhile, we also try to implement the RaceID¹² tool for rare cell identification (details shown in below; our response to comment #5). However, we do not identify meaningful rare cells in our single-cell data.

One major discovery of this work is the presence of COL1A1⁺/COL1A2⁺/IGFBP2⁺ fibroblasts in brain metastatic tissues that have not been previously reported. Considering the sample size and the limited number of single-cell data, identifying rare cell types is not a major focus of this work. Meanwhile, this work will contribute our in-house data to the growing brain metastatic scRNA-seq repository. Once such repository is large enough, it is more realistic for characterizing rare cell populations.

5. I also recommend authors try some of the computational tools for rare cell identification using scRNA seq (FIRE? CellSIUS?), as this might bring further novel insights to their work.

Response: Thanks for the reviewer's comments. We have tried to use computational tools including RaceID¹² to identify rare cells in our single-cell data.

Then following the tutorial of RaceID (<https://cran.r-project.org/web/packages/RaceID/vignettes/RaceID.html>), we first inspected the inferred cluster

number in the saturation plot shown below, which showed the decrease of the average within-cluster dispersion with increasing cluster number. The constant decrease indicated that the saturation was reached. The saturation plot below indicated that $k=4$ was the best cluster number for RaceID.

Moreover, we have also assessed the cluster stability by Jaccard's similarity, which was shown as below. Consistently, $k=4$ was the best cluster number for RaceID.

Then following the tutorial of RaceID (<https://cran.r-project.org/web/packages/RaceID/vignettes/RaceID.html>), we visualized the cell clusters identified by RaceID.

From the above figure, we found that RaceID did not identify rare cells. In contrast, it split tumor cell and NSC cell population into two clusters, each of them was not a rare cell population. Overall, the RaceID analysis concluded that rare cells were not detected in our single-cell data. As we mentioned before, one possible reason may be due to the limited number of cells. We will revisit this topic once more scRNA-seq data from brain metastasis tissues becomes available.

6. Could authors detail the statistics used for the Survival analysis? Can they share their code on GitHub for further review? In Fig 4 panel E, both the populations seem rather overlap in the survival curve. Why authors have used Average for patient stratification? They might use the median expression as a cutoff, or there are other statistically robust approaches for this. Please refer to this tutorial: <https://bioconnector.github.io/workshops/r-survival.html> [bioconnector.github.io]

Response: Thanks for the reviewer's comments. In our revised manuscript, we have modified using the median expression as the cutoff, and then did the 5-year survival analysis. The analysis codes were shared in our GitHub (<https://github.com/QSong-github/BrMet>) for review. The updated figures were shown in our revised Fig. 4D and Fig. 4E.

References

- 1 Abdulhaleem, M. *et al.* Collagen deposition within brain metastases is associated with leptomeningeal failure after cavity-directed radiosurgery. *Neurooncol Adv* **5**, vdac186 (2023). <https://doi.org:10.1093/noajnl/vdac186>
- 2 Gonzalez, H. *et al.* Cellular architecture of human brain metastases. *Cell* **185**, 729-745.e720 (2022). <https://doi.org:10.1016/j.cell.2021.12.043>
- 3 Jin, S. *et al.* Inference and analysis of cell-cell communication using CellChat. *Nature Communications* **12**, 1088 (2021). <https://doi.org:10.1038/s41467-021-21246-9>
- 4 Pasquini, G., Rojo Arias, J. E., Schäfer, P. & Busskamp, V. Automated methods for cell type annotation on scRNA-seq data. *Computational and Structural Biotechnology Journal* **19**, 961-969 (2021). <https://doi.org:https://doi.org/10.1016/j.csbj.2021.01.015>

- 5 Shao, X. *et al.* scCATCH: Automatic Annotation on Cell Types of Clusters from Single-Cell RNA Sequencing Data. *iScience* **23**, 100882 (2020). <https://doi.org/10.1016/j.isci.2020.100882>
- 6 Ianevski, A., Giri, A. K. & Aittokallio, T. Fully-automated and ultra-fast cell-type identification using specific marker combinations from single-cell transcriptomic data. *Nature Communications* **13**, 1246 (2022). <https://doi.org/10.1038/s41467-022-28803-w>
- 7 Aran, D. *et al.* Reference-based analysis of lung single-cell sequencing reveals a transitional profibrotic macrophage. *Nature Immunology* **20**, 163-172 (2019). <https://doi.org/10.1038/s41590-018-0276-y>
- 8 Zou, Y. *et al.* The Single-Cell Landscape of Intratumoral Heterogeneity and The Immunosuppressive Microenvironment in Liver and Brain Metastases of Breast Cancer. *Adv Sci (Weinh)* **10**, e2203699 (2023). <https://doi.org/10.1002/adv.202203699>
- 9 Stewart, C. A. *et al.* Single-cell analyses reveal increased intratumoral heterogeneity after the onset of therapy resistance in small-cell lung cancer. *Nature Cancer* **1**, 423-436 (2020). <https://doi.org/10.1038/s43018-019-0020-z>
- 10 Wu, F. *et al.* Single-cell profiling of tumor heterogeneity and the microenvironment in advanced non-small cell lung cancer. *Nature Communications* **12**, 2540 (2021). <https://doi.org/10.1038/s41467-021-22801-0>
- 11 Butler, A., Hoffman, P., Smibert, P., Papalexi, E. & Satija, R. Integrating single-cell transcriptomic data across different conditions, technologies, and species. *Nature Biotechnology* **36**, 411-420 (2018). <https://doi.org/10.1038/nbt.4096>
- 12 Grün, D. *et al.* Single-cell messenger RNA sequencing reveals rare intestinal cell types. *Nature* **525**, 251-255 (2015). <https://doi.org/10.1038/nature14966>

REVIEWERS' COMMENTS:

Reviewer #1 (Remarks to the Author):

The authors have addressed my concerns.

Reviewer #2 (Remarks to the Author):

The authors have addressed all my concerns.